# Implementation of mobile-health technology is associated with five-year survival among individuals in rural areas of Indonesia

**Asri Maharani** [1]*, **Sujarwoto**[2], **Devarsetty Praveen**[3], **Delvac Oceandy**[4,5],
**Gindo Tampubolon**[6], **Anushka Patel**[7]

**1** Division of Nursing, Midwifery and Social Work, The University of Manchester, Manchester, United Kingdom, **2** Department of Public Administration, University of Brawijaya, Malang, Indonesia, **3** The George Institute for Global Health, University of New South Wales, Hyderabad, India, **4** Division of Cardiovascular Sciences, The University of Manchester, Manchester Academic Health Science Centre, Manchester, United Kingdom, **5** Department of Biomedicine, Faculty of Medicine, University of Airlangga, Surabaya, Indonesia, **6** Global Development Institute, and NIHR Policy Research Unit on Older people and frailty, The University of Manchester, Manchester, United Kingdom, **7** The George Institute for Global Health, University of New South Wales, Sydney, Australia

\* asri.maharani@manchester.ac.uk

**Data Availability Statement:** The custodians of the data are the Malang District Health Agency (mortality) and The George Institute for Global

## Abstract

There is an urgent need to focus on implementing cost-effective health interventions and policies to reduce the burden of cardiovascular disease in Indonesia. This study aims to evaluate whether a mobile technology–supported primary health care intervention, compared with usual care, would reduce the risk of all-cause mortality among people in rural Indonesia. Data were collected from 11,098 participants in four intervention villages and 10,981 participants in four control villages in Malang district, Indonesia. The baseline data were collected in 2016. All the participants were followed for five years, and the mortality data were recorded. Cox proportional hazard model was used to examine the association between the intervention and the risk of all-cause mortality, adjusted for the covariates, including age, gender, educational attainment, employment and marital status, obesity and the presence of diabetes mellitus. During the five-year follow-up, 275 participants died in intervention villages, compared with 362 in control villages. Participants residing in intervention villages were at 18% (95%CI = 4 to 30) lower risk of all-cause mortality. Higher education attainment and being married are associated with lower risks of all-cause mortality among respondents who lived in the control villages, but not among those living in the intervention villages. A mobile technology–supported primary health care intervention had the potential to improve the five-year survival among people living in villages in an upper-middle income country.

## Author summary

Mobile technology in healthcare has been widely used in Indonesia. SMARThealth (Systematic Medical Appraisal Referral Treatment) is among them, and it has been piloted

Health (other data). Data and relevant meta-data will be made available to academic and other researchers after approval of a Data Access Request, subject to privacy and informed consent provisions, and in the case of mortality data, written approval from the data custodian. A request form can be obtained by email to the corresponding author or to DSC@georgeinstitute. org.

**Funding:** SMARThealth Indonesia was funded by a grant from Give2Asia (https://give2asia.org/) on the recommendation of the Pfizer Foundation (AP, DP) and the Australian National Health and Medical Research (NHMRC) program (https://www.nhmrc. gov.au/) grant APP1052555 (AP, DP, GT, DO, AM, SJ). The funders had no role in study design, data collection and analysis, decision to publish, or preparation of the manuscript.

**Competing interests:** The authors have declared that no competing interests exist.

and implemented in the Malang district of East Java province, Indonesia, since 2016. SMARThealth is a mobile device-based clinical decision support system (CDSS) to help primary healthcare workers improve optimal preventive treatment in primary healthcare. This paper evaluates whether this intervention would reduce the risk of death among people in Malang district. During the five-year follow-up, we found that participants residing in intervention villages had a lower risk of death than control villages. The findings highlight the potential of a mobile technology-based primary care intervention to empower primary healthcare workers to increase the population's survival in Malang district, Indonesia.

## Introduction

Globally, cardiovascular diseases (CVDs) were the cause of 17.9 million deaths in 2019, and more than three-quarters of them occurred in low- and middle-income countries (LMICs) [1,2]. There is a pressing need to focus on implementing cost-effective health interventions and policies to reduce the CVDs burden in those countries. Indonesia, the fourth most populated country globally, is among the LMICs facing the increasing burden of CVDs. In 2016, approximately one-third of all deaths in the country were caused by CVDs [3]. The prevalence of CVD risk factors, including smoking, physical activity, obesity, and hypertension, ranged between 28% and 33% among Indonesians aged 10 years and older in 2018 [4]. That prevalence varied among districts, with a higher prevalence of smoking and physical inactivity in rural areas.

Indonesia has a mixed public and private health system, which has decentralised healthcare services, including primary health care, to local district health agencies [5]. A study revealed service gaps across the country, particularly in rural areas, and showed that less than one-third of Indonesians with moderate to high risk of CVD received appropriate preventative treatments [6]. Among the efforts to reduce those unmet needs, the national government Ministry of Health organises and leads health promotion activities to raise public awareness, early screening and early detection of non-communicable diseases (NCDs) [5]. These activities are mainly held in community engagement programmes known as Integrated Health Post for Non-Communicable Diseases (*Pos Pembinaan Terpadu Penyakit Tidak Menular* or *Posbindu-PTM)*. *Posbindu-PTM* addresses NCD risk factors and is integrated into other settings within the community, such as schools and residences. *Posbindu-PTM* is mainly carried out by non-physician village healthcare volunteers (known as *kaders*).

To improve the provision of guideline-based assessment and management of CVD risk in Indonesia, we implemented an intervention known as SMART (Systematic Medical Appraisal Referral and Treatment) health in four villages in Malang District [7]. SMARThealth is a multifaceted technology-based primary care intervention that enables *kaders*, nurses at the village health posts and physicians at the primary healthcare centres to collaborate in assessing an adult's CVD risk using basic equipment and a clinical decision support app on a mobile device [7,8]. SMARThealth was implemented and evaluated in Indonesia through a study called SMARTHealth Extend. SMARThealth Extend was a quasi-experimental study that explored the potential of the intervention to identify and improve the care of people living with and at high risk of CVDs. The intervention was associated with significantly greater use of blood pressure-lowering medication (57% vs 16%) and a reduction in systolic blood pressure (-8.3 mmHg). We further found the intervention cost-effective to expand healthcare coverage to a

rural Indonesian population [9]. Despite these promising findings, the potential impact of the intervention on mortality is unknown.

As the intervention was carried out in rural areas with limited healthcare services and a high prevalence of CVDs, the estimated reductions in mortality are, therefore, particularly important to guide the policy-makers looking to implement interventions at scale. This study aimed to evaluate the association between the intervention on cardiovascular and all-cause mortality, using follow-up mortality data of the study cohorts.

## Materials and methods

### Study population and setting

We used longitudinal data (August 2016 – July 2021) from respondents recruited to SMARThealth Extend study in four interventions (Karangduren, Kepanjen, Sepanjang and Sidorahayu) and four control villages (Kendalpayak, Cepokomulyo, Mendalanwangi, and Majangtengah) in the Malang district of East Java province, Indonesia [7,9]. Baseline measurements were collected in August 2016 on 11,095 and 10,981 noninstitutionalised adults aged 40 and older in intervention and control villages, respectively. No sampling was involved in this study, as all eligible residents were invited to participate through complete household visits.

Assuming a baseline rate of 10%, a cluster size of 144 individuals, and an intraclass correlation coefficient of 0.05, eight villages that were equally divided between the intervention and control groups were estimated to provide 80% power with a 2-sided $\alpha$ =.05 to detect an absolute difference of 18% in the proportion of high-risk individuals taking appropriate preventive medications.

Covering an area of 3,535 square kilometres, Malang is the second largest district in the East Java province. In 2017, Malang had a population of 2.5 million distributed across 33 subdistricts, 390 villages, and 3,125 community neighbourhoods. The main economic source of the Malang district is agriculture, with an emphasis on rice and sugar cane. We selected the four villages in Malang because they represent rurality (urban, semi-urban, and rural villages), the occupation of most residents, proximity to a tobacco factory, and population density. Kepanjen and Karangduren represented urban villages, while Sidorahayu and Sepanjang represented semi-urban and rural villages. All of those villages had sugar cane plantations in their areas. Four control villages were subsequently chosen as a match with each intervention village based on population size, rurality, predominant occupation of residents, proximity to a tobacco factory, and the number of health workers.

### Data collection

Trained enumerators administered tablet-based questionnaires to collect sociodemographic, physical activity level, medical history, height and weight. We entered age as a continuous variable (years) and sex as a binary variable with males as the reference. We classified the levels of education completed by respondents as less than senior high school (reference) and senior high school or higher degree. Marital status was classified into married and not married (reference). Employment status was included in this study, with employed as the reference. We calculated Body Mass Index (BMI) as body weight (kg) divided by height squared (m2). We then categorised the respondents as obese if their BMI was $\geq$ 30 kg/m2.

A random blood glucose sample was collected (with a record of the time since last ate) using the pin-prick method and portable glucometers (FreeStyle Optium Neo) [10]. We defined respondents as previously diagnosed diabetes by a "yes" answer to the question, "Has a doctor/other health professionals ever told you that you had diabetes?" These individuals were asked to provide additional information about the current use of any medications for diabetes.

Respondents were classified as having diabetes mellitus if they reported having physician-diagnosed diabetes mellitus and were either taking hypoglycaemic medication or had FPG$\geq$ 7.0 mmol/L or 2-hour PG $\geq$11.1 mmol/L.

## Mortality data

A follow-up survey was done in 2021. The Malang health authority independently conducted the survey as part of the annual mortality census in the district. Out of 22,103 individuals in the initial survey, 637 had died, and 17 were lost at follow-up (0.07%). Information on death was obtained from the study participant's family members. The proportion of deaths with certification was 74.6% (475 participants); the rest was ascertained using verbal autopsy [11,12]. Trained health officers carried out the verbal autopsy. Death certificates were completed by medical doctors in hospitals if the patients died in hospitals or primary care centres if the patients died at home. Deaths in the cohort that occurred between August 2016 and July 2021 were recorded. The primary endpoint for this study was all-cause mortality.

## Ethics approval

The study received ethics approval from the Ethical Committee, Ministry of Research, Technology, and Higher Education, Medical Faculty of Brawijaya University (330/EC/KEPK/08/ 2016) and was registered on the Clinical Trial Registry of India (CTRI/2017/08/009387). Ethics approval to use the mortality data was granted from Brawijaya University Number 124-KEP-UB-2021 based on an approval letter from Malang Health Authority Number 813/ 6679/3507103/2021. Written informed consent was obtained from all participants contributing data to the analyses. Patients or the public were not involved in our research's design, conduct, reporting, or dissemination plans.

## Statistical analyses

We summarised the categorical baseline characteristics as frequencies and percentages, while continuous variables were summarised using means and standard deviations. Using the chi-square test and Kruskal Wallis for categorical and continuous variables, we then compared the baseline characteristics between respondents by their mortality status in the intervention and control villages.

To examine the association between SMARThealth intervention and all-cause mortality, we employed the Cox proportional hazard model. The proportional hazard assumption was examined by plotting the Kaplan-Meier survival curves and testing the interaction between residing in intervention villages and follow-up time. The reference group in this study was respondents residing in control villages. Hazard ratios were adjusted for age, sex, education attainment, marital status, employment status, physical activity, and the presence of obesity and diabetes mellitus. We performed the analysis of two models. In the first model, we included all respondents and examined the relationships between living in the intervention or control villages and all-cause mortality. The second model performed the analysis separately for intervention and control villages.

The nature of this quasi-randomised experiment means that some dependence between intervention and control villages is unavoidable. It is possible that non-random baseline differences in mortality existed despite our effort to match them on population size, rurality, predominant occupation of residents, proximity to a tobacco factory, and number of health workers. We used the weighting scheme to address these differences, with weights proportional to the inverse probability of receiving intervention. The covariates include the deprivation index of each village.

We performed four sensitivity analyses. Firstly, we performed the Cox proportional hazard model to identify the effect of SMARThealth intervention on CVD mortality. By the end of this follow-up period (five years), 421 deaths due to CVD causes and 216 deaths due to non-CVD causes occurred. Second, we performed competing-risk regression analysis with subdistribution hazard ratios (SHR) and related 95% Confidence Intervals (CIs), using a version of the Fine and Gray method [13] to identify the association between SMARThealth intervention and CVD mortality as a potential competing risk. This method allows a competing risk – an event that might occur during the follow-up instead of the event of interest – to be considered in the analysis. In this case, non-CVD mortality is a potential competing risk when examining CVD mortality; therefore, it is important to take this into account, rather than treating those who had died due to non-CVD causes as censored. Secondly, we performed the analysis with the follow-up until 2 March 2020 to understand the effect of the COVID-19 pandemic on our study. Finally, we performed the analysis only for respondents with death certificates. All analyses were performed using STATA software, version 17.

## Role of the funding source

The funder of this study had no role in the study design, data collection, data analysis, data interpretation, or writing of the report. The corresponding author had full access to all the data in the study and had final responsibility for the decision to submit for publication.

## Results

The mean age of participants in the intervention and control villages at baseline was 54.7 (SD 10.6) and 55.1 (SD 10.9) years, respectively (Table 1). The proportion of females in the intervention villages (57.9%) was slightly higher than that in the control villages (55.3%). Approximately 35% of the respondents were unemployed in the intervention, while the proportion of unemployment in the control villages was 32.1%. The mean BMI among total respondents was 25.3 (SD=6.3) kg/m2 and 25.1 (SD=5.9) kg/m2 in the intervention and control villages, respectively.

After 5 years of follow-up, 275 respondents in intervention villages and 362 respondents in control villages died. Respondents who died in both intervention and control villages were likely to be older, had lower educational attainment, were less physically active and had

**Table 1. Baseline characteristics of study participants by mortality.**

| | | Intervention villages | | | | Control villages | | |
|---|---|---|---|---|---|---|---|---|
| | Total (n=11,098) | Non death (n=10,823) | Death (n=275) | P value* | Total (n=10,981) | Non death (n=10,619) | Death (n=362) | P value* |
| Age, mean (sd) | 54.71 (10.57) | 54.50 (10.46) | 62.47 (11.57) | <0.001 | 55.09 (10.94) | 54.80 (10.73) | 63.60 (13.46) | <0.001 |
| Female, n (%) | 6,426 (57.92) | 6,288 (58.11) | 138 (50.18) | 0.009 | 6,075 (55.32) | 5,903 (55.59) | 172 (47.51) | 0.002 |
| Senior high school or higher degree, % | 2,990 (26.95) | 2,939 (27.16) | 51 (18.55) | 0.006 | 2,286 (20.82) | 2,246 (21.15) | 40 (11.05) | <0.001 |
| Married, % | 9,012 (81.23) | 8,806 (81.39) | 206 (74.91) | 0.007 | 8,968 (81.67) | 8,729 (82.20) | 239 (66.02) | <0.001 |
| Unemployed, % | 3,880 (34.97) | 3,751 (34.67) | 129 (46.91) | <0.001 | 3,584 (32.64) | 3,410 (32.11) | 174 (48.07) | <0.001 |
| Vigorous physical activity, % | 2,915 (26.26) | 2,863 (26.46) | 52 (18.91) | 0.005 | 2,701 (24.60) | 2,635 (24.81) | 66 (18.23) | 0.004 |
| BMI, mean (sd) | 25.35 (6.27) | 25.38 (6.23) | 24.11 (7.61) | <0.001 | 25.07 (5.93) | 25.12 (5.75) | 23.61 (9.80) | <0.001 |
| Obese, % | 1,601 (14.50) | 1,563 (14.51) | 38 (14.13) | 0.859 | 1,479 (13.55) | 1,449 (13.72) | 30 (8.47) | 0.005 |
| Diabetes, % | 931 (8.39) | 886 (8.19) | 45 (16.36) | <0.001 | 715 (6.51) | 675 (6.36) | 40 (11.05) | <0.001 |

**Note:** The bivariate analysis was performed using the chi-square test for categorical and Kruskal Wallis for continuous variables

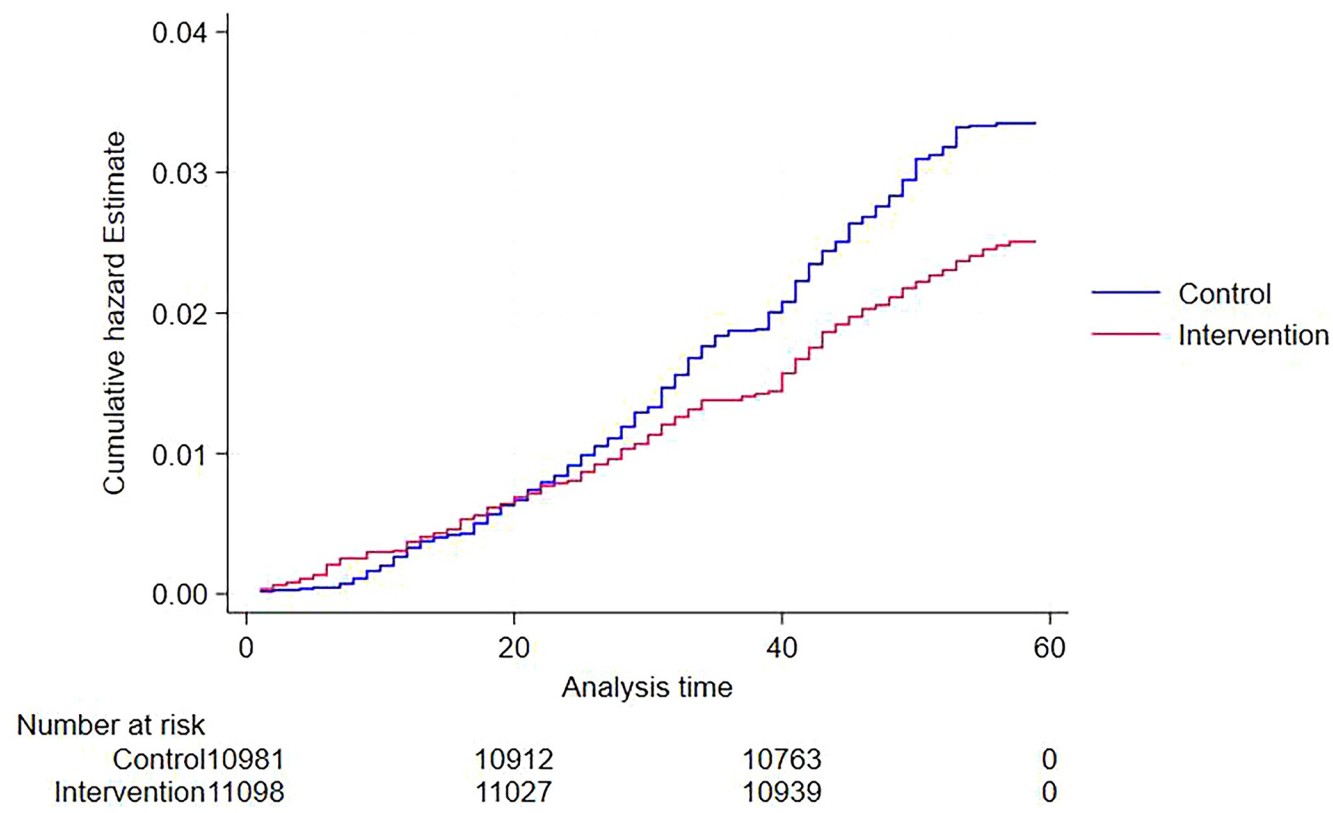

**Fig 1. Estimates of the cumulative incidence curves of all-cause mortality in intervention and control villages.**

diabetes. The Kaplan-Meier curve (Fig 1) for the unadjusted rate of all-cause mortality shows differences in risk according to the presence of the intervention. Fig 2 shows the estimated hazard ratio and 95% confidence intervals for all-cause mortality according to the presence of intervention. It shows that the participants who lived in the intervention villages had a lower risk of all-cause mortality (HR=0.82; 95%CI=0.70 to 0.96) than those in the control villages.

Fig 3 shows separate Cox proportional hazard models for intervention and control villages. Higher education and marriage were associated with lower all-cause mortality in the control villages, but not in the intervention villages. Being unemployed was related to higher all-cause mortality risk among respondents living in the control village, but not in the intervention villages. In the intervention villages, respondents with diabetes mellitus had twice the higher hazard ratios of all-cause mortality (adjusted HR=2.12; 95% CIs=1.52 to 2.97). The presence of diabetes mellitus was also a significant risk factor for all-cause mortality (HR=1.79; 95% CIs=1.27 to 2.53) in the control villages.

S1 Table and S2 Table describe the baseline characteristics of respondents by the cause of mortality. Of the 11,095 respondents in the intervention villages, 169 (1.5%) and 106 (0.9%) died due to CVD and non-CVD causes, respectively, after five years of follow-up. The percentages of respondents who died due to CVD and non-CVD causes in the control villages were 2.3% for CVD and 1.0% for non-CVD mortality.

The first sensitivity analysis (S1 Fig) shows the results of Cox proportional hazard models to identify the link between SMARThealth intervention and the risk of CVD mortality. It shows that the participants who lived in the intervention villages had a lower risk of CVD mortality (SHR=0.74; 95%CI=0.60 to 0.90). The Cox proportional hazard models were performed separately for intervention and control villages in S2 Fig.

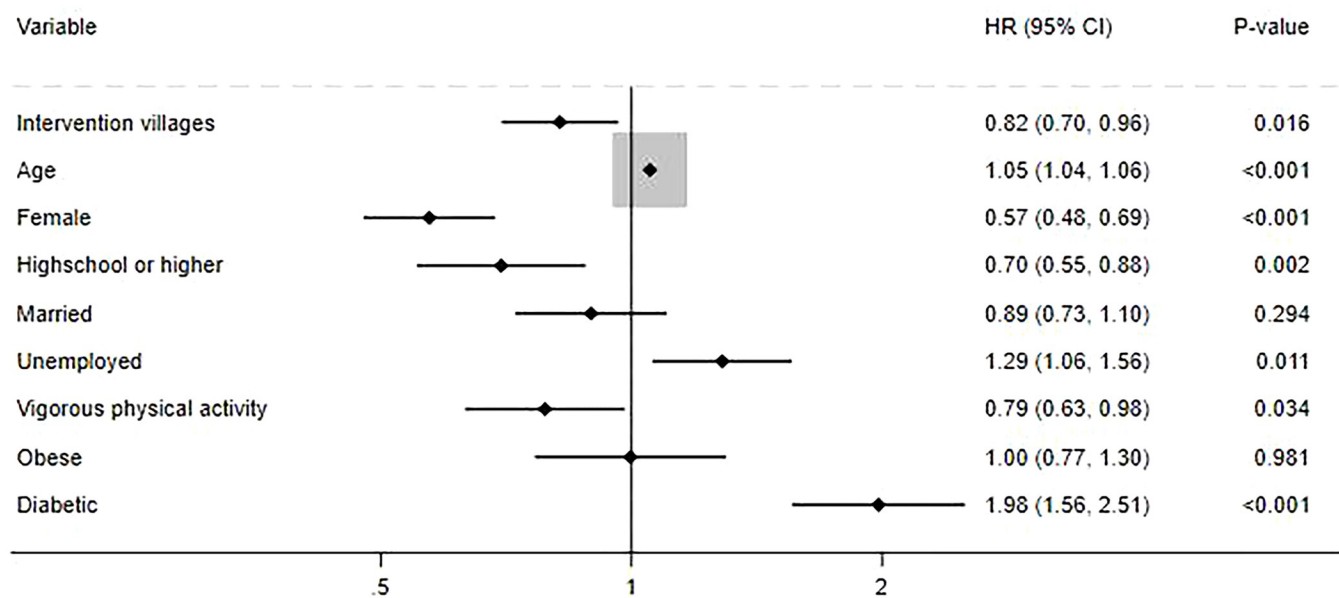

**Fig 2. Cox proportional hazard models predicting the risk of all-cause mortality.**

S3 Fig shows that participants residing in intervention villages had 29 (95% CIs=14 to 42) % lower SHR of CVD mortality than in control villages. After adjustment for all covariates, the association between the intervention and the risk of CVD mortality remains significant (adjusted SHRs=0.74; 95% CIs=0.61 to 0.90).

The adjusted models show that older age was associated with a 1.05 (95% CIs=1.04-1.06) higher risk of CVD mortality. Female gender (adjusted SHRs=0.64; 95% CIs=0.51 to 0.79) was associated with a lower risk of death due to CVD causes. Having vigorous physical activity was related to 27 (95% CIs=4 to 45%) lower risk of CVD mortality, and respondents with diabetes mellitus had a 1.95 (95%CI=1.45 to 2.61) higher risk of CVD mortality after five years in the adjusted model.

We truncated the follow-up on 2 March 2020 in S4 Fig. It shows that the participants who lived in the intervention villages had a lower risk of CVD mortality (SHR=0.75; 95%CI=0.61 to 0.91), indicating that the COVID-19 pandemic may not affect the relationships between

**A. Intervention villages**

**B. Control villages**

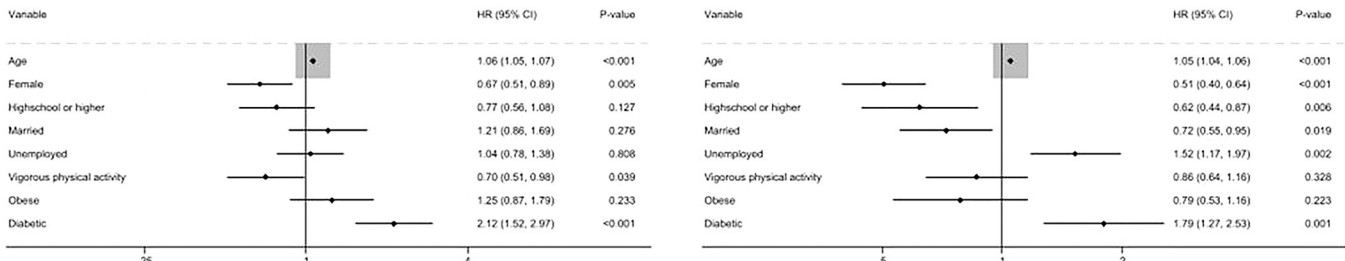

**Fig 3. Cox proportional hazard models predicting the risk of all-cause mortality in (A) intervention and (B) control villages.**

SMARThealth and CVD mortality in this study. **S5 Fig** shows that the participants with death certificates who lived in the intervention villages had a lower risk of CVD mortality (SHR=0.67; 95%CI=0.53 to 0.84) than those who lived in the control villages. This result indicates that the methods used for determining the cause of death may not affect the effect of SMARThealth intervention on the risk of CVD mortality in this study.

## Discussion

In the present study, we found that the implementation of a complex technology-enabled primary care intervention was associated with lower all-cause deaths in a rural Indonesian population over the subsequent five years. Individuals who received the intervention have 18% lower risk of all-cause mortality than those living in control villages. The findings show the potential of a technology-supported intervention to empower community health workers to increase the survival of the population in rural areas in an LMIC. In the Indonesian context, the community health workers provide health education and screening for non-communicable disease risk factors, mainly blood pressure, blood glucose and cholesterol, for the community. The magnitude of benefit in mortality reduction is proportional to the benefits observed in improved blood pressure control, a difference of 41% [7]. The estimated difference in mortality of 32% seems entirely proportional. Our finding suggests that the methods used to determine the cause of death did not affect the relationship between the SMARThealth intervention and the risk of CVD mortality (**S5 Fig**). A prior study showed that participation in the activities provided by community health workers was related to 50% higher odds of being aware and 118% higher odds of receiving treatment among individuals with hypertension in rural Indonesia [14]. The SMARThealth intervention was related to higher awareness and use of BP-lowering medication [7]. As community health workers screen the CVD risks and promote healthy behaviours in the SMARThealth intervention, the participants living in the intervention villages might have higher awareness and treatment for CVDs, improving their survival.

We further found that female gender, educational attainment and employment status were independently associated with the risk of all-cause mortality among respondents who did not receive SMARThealth interventions but not among those who lived in the intervention villages. Prior studies revealed the association between the female gender [15,16] and higher education attainment [17] with lower CVD mortality risks. *Kaders* screened 86.4% of the census population in the intervention villages through household visits in the SMARThealth Extend intervention [7]. In addition, they gave health promotion through several means for the screened populations in the intervention villages, including videos embedded in the SMARThealth app. Our findings show that the intervention may reduce the effect of gender and socio-economic factors on CVD mortality. The intervention is more likely to enable health services and information to reach those with lower education and socio-economic status and males.

The presence of diabetes mellitus was associated with a higher risk of CVD mortality in both intervention and control villages. Those findings were consistent with previous observations in low, middle and high-income countries [18–23]. This study further adds that the presence of diabetes mellitus was associated independently with a reduced life span, ranging from 1.5 years to 2.6 years among those aged 40 to 80 years old. Our findings support the important role of diabetes mellitus in mortality risk. Given the high prevalence of diabetes and the increased number of years of exposure to diabetes in Indonesia, the diversified causes of mortality could have important clinical and public health effects for diabetes in coming decades, indicating a need to identify and emphasise prevention approaches and epidemiological monitoring of a wider range of diabetes-related morbidity.

This study has several strengths. Although not nationally representative, the large sample size, diversity of areas covered, and broadly consistent findings across study population subgroups mean that the present hazard risk estimates are likely not biased and can be generalisable to the population at large. The other strengths of this study include standardised approaches, stringent quality control for data collection, and completeness of follow-up.

### Research limitations

However, the study also has several limitations. The first limitation is the unavailability of fasting blood glucose measurements from study participants. We only collected the data on fasting blood glucose if the respondents fasted before the data collection. Otherwise, we collect the random blood glucose. Another limitation is that some residual confounding might exist, although we used a wide range of confounders. For instance, we could not adjust the dietary pattern [24] and cholesterol level [25]. Finally, around a quarter of the deaths of the respondents were ascertained using verbal autopsy. The determination of whether the death received verbal autopsy or not is unlikely related to the risk factors or cause of death; hence, it is unlikely to be biased. Instead, it relates to funding available to the district health office. No physical autopsies were performed on the participants who died. Prior studies showed that verbal autopsy protocols in the Indonesian Sample Registration System provide sufficient quality evidence on the underlying cause of death [11,12]. Also, there are no better alternatives for rural populations long denied access to basic CVD diagnosis, as our prior study found that around two-thirds of individuals with moderate to high-risk CVD failed to receive preventative treatments [6]. More routine use of verbal autopsies thus should be considered when mortality information is severely lacking in Indonesia. To test the difference in the proportion of verbal autopsies in intervention and control villages, we performed a prtest in STATA.17 and found that there is no significant difference in the proportion of verbal autopsies between the two village groups. We have further performed the sensitivity analysis, including only respondents with death certificates, and the significant effect of SMARThealth intervention on the lower CVD mortality remains.

### Conclusions

Our findings add to the literature by showing the potential for technology-enabled primary care interventions to reduce the risk of all-cause and CVD mortality in a rural Indonesian population. However, this conclusion could be limited, given the observational nature of the analyses. This result indicates the potential of similar interventions to strengthen health systems and improve the life span of the population in rural and resource-constrained environments.

### Supporting information

**S1 Table. Baseline characteristics of study participants.**
(DOCX)

**S2 Table. Baseline characteristics of study participants in intervention and control villages.**
(DOCX)

**S1 Fig. Cox proportional hazard models predicting the risk of CVD mortality.**
(TIF)

**S2 Fig.** Cox proportional hazard models predicting the risk of CVD mortality in (A) intervention and (B) control villages.
(PDF)

**S3 Fig.** Subdistribution hazard ratios (95 CIs) for the association between intervention and cardiovascular mortality: (A) Unadjusted Model and (B) Adjusted Model.
(PDF)

**S4 Fig.** Fine and Gray subdistribution hazard of cardiovascular mortality in (A) all, (B) intervention and (C) control with the follow-up truncated on 2 March 2020.
(PDF)

**S5 Fig. Fine and Gray subdistribution hazard of cardiovascular mortality in all villages among respondents with death certificate.**
(TIF)

## Author Contributions

**Conceptualization:** Asri Maharani, Sujarwoto, Gindo Tampubolon, Anushka Patel.

**Data curation:** Asri Maharani.

**Formal analysis:** Asri Maharani.

**Investigation:** Asri Maharani.

**Methodology:** Asri Maharani, Sujarwoto, Devarsetty Praveen, Delvac Oceandy, Gindo Tampubolon, Anushka Patel.

**Project administration:** Sujarwoto.

**Resources:** Sujarwoto.

**Visualization:** Asri Maharani.

**Writing – original draft:** Asri Maharani, Sujarwoto.

**Writing – review & editing:** Asri Maharani, Sujarwoto, Devarsetty Praveen, Delvac Oceandy, Gindo Tampubolon, Anushka Patel.

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
