## [Decision Letter · Decision Letter 0]

13 Nov 2023

PDIG-D-23-00342

Implementation of mobile-health technology is associated with lower all-cause mortality after 5 years of follow-up in rural areas of Indonesia

PLOS Digital Health

Dear Dr. Maharani,

Thank you for submitting your manuscript to PLOS Digital Health. After careful consideration, we feel that it has merit but does not fully meet PLOS Digital Health's publication criteria as it currently stands. Therefore, we invite you to submit a revised version of the manuscript that addresses the points raised during the review process.

Please submit your revised manuscript within 60 days Jan 12 2024 11:59PM. If you will need more time than this to complete your revisions, please reply to this message or contact the journal office at digitalhealth@plos.org. Please include the following items when submitting your revised manuscript:

We look forward to receiving your revised manuscript.

Kind regards,

Haleh Ayatollahi

Section Editor

PLOS Digital Health

Journal Requirements:

Additional Editor Comments (if provided):

Reviewers' comments:

Reviewer's Responses to Questions

**Comments to the Author**

1. Does this manuscript meet PLOS Digital Health’s publication criteria? Is the manuscript technically sound, and do the data support the conclusions? The manuscript must describe methodologically and ethically rigorous research with conclusions that are appropriately drawn based on the data presented.

Reviewer #1: Yes

Reviewer #2: Yes

Reviewer #3: Yes

2. Has the statistical analysis been performed appropriately and rigorously?

Reviewer #1: Yes

Reviewer #2: I don't know

Reviewer #3: Yes

3. Have the authors made all data underlying the findings in their manuscript fully available (please refer to the Data Availability Statement at the start of the manuscript PDF file)?

Reviewer #1: Yes

Reviewer #2: No

Reviewer #3: Yes

4. Is the manuscript presented in an intelligible fashion and written in standard English?

Reviewer #1: Yes

Reviewer #2: Yes

Reviewer #3: Yes

5. Review Comments to the Author

Reviewer #1: PDIG-D-23-00342

Clarify the data availability statement. I suspect you mean that confidential data is available through the George Institute, and that you will deposit the de-identified data in a repository. Right now, it’s not clear. You will need to provide the link to the publicly available repository.

Abstract: 

The background should include a sentence from your introduction explaining the importance of the issue, e.g., “There is a pressing need to focus on implementing cost-effective health interventions and policies to reduce the cardiovascular disease burden in Indonesia.”

In the findings, the description of results is a little confusing, you probably meant to say something like this: “Participants residing in intervention villages were at 18% (95%CI = 4 to 30) lower risk of all-cause mortality.”

Introduction:

In the last sentence of the first paragraph, I think there should be the word “and” between smoking and physical activity.

I was confused about the distinction between SMARThealth Extend and SMARThealth. Please describe these in more detail.

Materials and methods:

Consider using more subheadings.

The sentence about the random blood sample (line 115) seems confusing. Consider taking out the words “were measured.”

Line 127, you say “Out of 22,103 individuals in the follow-up…” But I’m wondering if you mean the initial survey, since later you mentioned 17 individuals lost to follow-up.

On line 133, the word “one” is confusing. I think you mean to say that the determination of whether the death was recorded via verbal autopsy or not is unrelated to the risk factors or cause of death. This and the following sentences may be better off in the discussion section.

On line 135, do you mean “No physical autopsies were performed on the participants who died.”?

Results:

Figures

All the figures need legends. For example, Figure 1 needs a legend to distinguish between the two colors on the graphed lines. Please label the x axis. The “number at risk” is a bit confusing. You might want to explain in the text or remind people in a footnote to the figure that the reason it is 0 at month 60 is because that is the end of the follow-up timeframe. Also, in the text, you use the term “hazard ratio,” but the tables embedded in the figures use OR, or Odds Ratio. Explain in the text or in a footnote to the figure that HR is equivalent to OR, or change the headings on the tables. Please explain more about the horizontal bars in the figures. They appear to be a graphical description of the Hazard Ratio, but there’s no explanation in the figure itself or in a footnote below the figure. Figures 2 and 3 appear a bit blurry.

Discussion:

Line 257, please explain more about how the use of verbal autopsies strengthened the study.

Lines 261 and 263, I was still a bit confused between SMARThealth Extend and SMARThealth intervention.

Line 269, I’m wondering if the word “association” would be better here than the word “effect.”

On line 276, add the word “status” after “socio-economic.”

On line 293, it might be clearer if you reworded to say “The first limitation is the unavailability of fasting blood glucose measurements from study participants.”

On line 300, please re-word this sentence. It’s currently unclear.

Reviewer #2: I appreciate the opportunity to analyze this manuscript and I find the idea of using low-cost technology to improve access to healthcare very interesting, especially when they show such an impact as reducing mortality.

I suggest some changes and corrections below (following the order in which they appear in the text):

1. I believe there is a small typing error that hinders understanding in line 53

2. in lines 61 and 62 two government programs are mentioned, I suggest putting the translation of the terms that, I believe, are in the Indonesian language

3. in "Materials and methods" it was mentioned that sampling was not carried out since all eligible residents were invited. Even so, I suggest that a sample calculation be carried out to correctly determine the power of the study.

4. I also missed a baseline table comparing each variable in the 2 groups (intervention and control) with p value for the difference between the groups. This was almost done in table 1, but there is no comparison between the groups, only within each group between death and non-death. I understand that these differences were controlled in the following analyses, but I still think it is important to have a table that summarizes the differences between the control and intervention groups.

Reviewer #3: The paper aims evaluate whether a mobile technology–supported primary health care intervention, compared with usual care, would reduce the risk of all-cause mortality among people in rural Indonesia.

The overall quality of the study is very commendable. 

The authors outlined the aims and methodology of the study.

The results were outlined and inferences from the data were clearly backed up with rigorous statistical analysis and robust discussions. 

It also meets the requisite ethical standards.

The study is relevant and of significance in that it uses technology to improve access to the rural areas.

However, my main observation is on stretching (attributing) the outcome of the use of the SMARThealth App to reduction in “All-cause mortality’ instead of “CVD-related mortality” since SMARThealth (intervention) is CVD focused. This is more so that other cofounders that could have influenced mortalities from non-CVD causes (which is about two thirds of the cause of mortalities in the country) were not adequately controlled for.

I think it would be more appropriate to have the title of the study as Implementations of mobile-health technology is associated with lower CVD-related mortality after 5 years of follow-up in rural areas of Indonesia and this aligns with the conclusion of the authors i.e. “In conclusion, our findings add to the literature by showing the potential technology-enabled primary care interventions in reducing the risk of CVD mortality in a rural Indonesian population (line 311-313)

It is however worth noting that changing the title does not necessarily change the interpretation that “ A mobile technology–supported primary health care intervention had the potential to improve the five-year survival among people in the rural areas in an upper-middle income country. 

The article is recommended for publication with proposed amendments.

6. PLOS authors have the option to publish the peer review history of their article (what does this mean?). If published, this will include your full peer review and any attached files.

**Do you want your identity to be public for this peer review?** For information about this choice, including consent withdrawal, please see our Privacy Policy.

Reviewer #1: No

Reviewer #2: No

Reviewer #3: No

---

## [Decision Letter · Decision Letter 1]

30 Jan 2024

PDIG-D-23-00342R1

Implementation of mobile-health technology is associated with lower all-cause mortality after 5 years of follow-up in rural areas of Indonesia

PLOS Digital Health

Dear Dr. Maharani,

Thank you for submitting your manuscript to PLOS Digital Health. After careful consideration, we feel that it has merit but does not fully meet PLOS Digital Health's publication criteria as it currently stands. Therefore, we invite you to submit a revised version of the manuscript that addresses the points raised during the review process.

Please submit your revised manuscript within 30 days Feb 29 2024 11:59PM. If you will need more time than this to complete your revisions, please reply to this message or contact the journal office at digitalhealth@plos.org. Please include the following items when submitting your revised manuscript:

We look forward to receiving your revised manuscript.

Kind regards,

Haleh Ayatollahi

Section Editor

PLOS Digital Health

Journal Requirements:

Additional Editor Comments (if provided):

I appreciate the authors for their time and efforts to revise the manuscript. There are a few minor issues which need further attentions. I suggest you to follow the journal instructions for preparing the abstract and choosing appropriate headings and subheadings for the manuscript. Please add the “Research limitation” and “Conclusion” as headings, too.

Reviewers' comments:

Reviewer's Responses to Questions

**Comments to the Author**

1. If the authors have adequately addressed your comments raised in a previous round of review and you feel that this manuscript is now acceptable for publication, you may indicate that here to bypass the “Comments to the Author” section, enter your conflict of interest statement in the “Confidential to Editor” section, and submit your "Accept" recommendation.

Reviewer #1: All comments have been addressed

Reviewer #2: All comments have been addressed

Reviewer #3: All comments have been addressed

2. Does this manuscript meet PLOS Digital Health’s publication criteria? Is the manuscript technically sound, and do the data support the conclusions? The manuscript must describe methodologically and ethically rigorous research with conclusions that are appropriately drawn based on the data presented.

Reviewer #1: Yes

Reviewer #2: Yes

Reviewer #3: Yes

3. Has the statistical analysis been performed appropriately and rigorously?

Reviewer #1: Yes

Reviewer #2: I don't know

Reviewer #3: Yes

4. Have the authors made all data underlying the findings in their manuscript fully available (please refer to the Data Availability Statement at the start of the manuscript PDF file)?

Reviewer #1: Yes

Reviewer #2: No

Reviewer #3: Yes

5. Is the manuscript presented in an intelligible fashion and written in standard English?

Reviewer #1: Yes

Reviewer #2: Yes

Reviewer #3: Yes

6. Review Comments to the Author

Reviewer #1: Check the requirements for the financial disclosure statement shown in the table on the first few pages prior to the manuscript. It looks like the Journal wants the funding statement to include author initials for who received the funding, the URL of the funder, and further details on the role of the funder in designing implementing and publishing the study results.

Reviewer #2: (No Response)

Reviewer #3: The author has satisfactorily addressed the feedback given.

7. PLOS authors have the option to publish the peer review history of their article (what does this mean?). If published, this will include your full peer review and any attached files.

**Do you want your identity to be public for this peer review?** For information about this choice, including consent withdrawal, please see our Privacy Policy. 

Reviewer #1: No

Reviewer #2: No

Reviewer #3: No

---

## [Editor Report · Decision Letter 2]

23 Feb 2024

Implementation of mobile-health technology is associated with lower all-cause mortality after 5 years of follow-up in rural areas of Indonesia

PDIG-D-23-00342R2

Dear Dr Maharani,

We are pleased to inform you that your manuscript 'Implementation of mobile-health technology is associated with lower all-cause mortality after 5 years of follow-up in rural areas of Indonesia' has been provisionally accepted for publication in PLOS Digital Health.

Best regards,

Haleh Ayatollahi

Section Editor

PLOS Digital Health